# A study about management of drugs for leprosy patients under medical monitoring: A solution based on AHP-Electre decision-making methods

Igor W. S. Falcão[1,2☯¶]*, Daniel S. Souza[1☯], Diego L. Cardoso[1☯], Fernando A. R. Costa[5☯], Karla T. F. Leite[3☯], Harold D. de M. Junior[4☯], Claudio G. Salgado[2☯‡], Moisés B. da Silva[2‡], Josafá G. Barreto[2‡], Patricia F. da Costa[2‡], Adriano M. dos Santos[1☯], Guilherme A. B. Conde[6‡], Marcos C. da R. Seruffo[1,2☯‡]

1 Technology Institute, Federal University of Para, Belém, PA, Brazil, 2 Dermato-Immunology Laboratory, Federal University of Para, Marituba, PA, Brazil, 3 Computer Science, Rio de Janeiro State University, Rio de Janeiro, RJ, Brazil, 4 Electrical Engineering Department, Rio de Janeiro State University, Rio de Janeiro, RJ, Brazil, 5 Center for Higher Amazon Studies, Federa University of Para, Belém, RJ, Brazil, 6 Institute of Engineering and Geosciences - IEG, Federal University of Western Para, Belém, PA, Brazil

☯ These authors contributed equally to this work.
‡ CGS, MBS, JGB, PFC, GABC, and MCRS also contributed equally to this work.
¶ Membership list can be found in the Acknowledgments section.
* igorufpa2013.4@gmail.com

**Data Availability Statement:** The data used in this object of study involve personal information of patient care in clinical treatment. Data are available

## Abstract

Leprosy, also known as Hansen's, is one of the listed neglected tropical diseases as a major health problem global. Treatment is one of the main alternatives, however, the scarcity of medication and its poor distribution are important factors that have driven the spread of the disease, leading to irreversible and multi-resistant complications. This paper uses a distribution methodology to optimize medication administration, taking into account the most relevant attributes for the epidemiological profile of patients and the deficit in treatment via Polychemotherapy. Multi-criteria Decision Methods were applied based on AHP-Electre model in a database with information from patients in the state of Para between 2015 and 2020. The results pointed out that 84% of individuals did not receive any treatment and, among these, the method obtained a gain in the distribution of 68% in patients with positive diagnosis for leprosy.

## Introduction

Leprosy is a chronic infectious disease caused by the *Mycobacterium leprae*. This is one of the diseases that endanger human health, with an incubation period that can take up to decades [1, 2]. The reflexes of the disease usually cause damage to the skin and nerves, causing immune disorders that can trigger inflammatory episodes [3]. There is currently no gold standard diagnostic test for Leprosy, a fact that often leads to irreversible disabilities for individuals [4].

on GitHub (https://github.com/igorfalcao/hansysLeprosy). Access to data is allowed for the entire academic community that intends to contribute positively to this research.

**Funding:** This work was supported by CNPq (486183/2013-0 CNPq grant for MS; 448741/2014-8 grant for JB; 428964/2016-8 grant and 313633/2018-5 fellowship for CS; and 306815/2018-4 grant for AR-d-S), CAPES PROAMAZONIA 3288/2013, CAPES Biocomputacional – RPGPH (3381/2013), Brazil Ministry of Health 035527/2017, PROPESP/UFPA, VALE S.A. 27756/2019, Fulbright Scholar to Brazil 2019-2020 (JS), and the Heiser Program of the New York Community Trust for Research in Leprosy (JB, MS, CS, and JS) grants P15-000827, P16-000796, and P18-000250. The funders had no role in study design, data collection, analysis, interpretation, or writing of the report.

**Competing interests:** No, the authors have declared that no competing interests exist.

In this context, the delay in diagnosis/treatment can lead to permanent deformities, such as peripheral nerve lesions and severe deformities, which, besides aggravating the condition of patients, intensifies the impacts of social stigmas (A sign that designates the bearer as disqualified) [5, 6]. Although the infection is responsive, disabilities are now irreparable in some cases [7]. This fact highlights the importance and need for early detection of leprosy.

According to World Health Organization (WHO, 2018) records, three countries reported more than 10,000 new cases of Leprosy, including India (120,334), Brazil (28,600) and Indonesia (17,017), representing 81% of the new cases detected worldwide [8]. In Brazil specifically, after 13 years of decrease in the amount of cases, the number of diagnosed patients increased again in 2017, with a prevalence of 4.44 cases of Leprosy/10,000 inhabitants only in the Midwest region [9].

WHO in 2016 launched a global strategy for Leprosy from 2016-2020 to further reduce the disease burden at the global and local level with three point targets: (a) Zero G2D (grade 2 disability) in children diagnosed with leprosy; (b) reduction of new leprosy cases with G2D to less than 1 per million population; and (c) zero countries with legislation allowing discrimination due to leprosy [10]. However, low adherence against the disease is still a significant obstacle in its control, as defaulters with incomplete cure remain as potential sources of infection [11].

Currently, the treatment of Leprosy is done via multidrug therapy, or MDT, which involves drug administration consisting of *rifampicin, clofazimine and dapsone* for a period of 6 to 24 months, depending on the type and grade of the disease. The treatment is made available by the Brazilian Federal Government (via the Ministry of Health) and partnerships with public institutions [12]. However, access to public services in Brazil and other developing countries can be difficult. There are records of patients who may wait more than a year to receive specialized evaluation, impairing prognosis. In addition, physicians may face difficulties in making the diagnosis in a primary care facility [13].

Leprosy is a difficult disease to diagnose, which has a wide range of symptoms as well as a high capacity for contagion [14]. These characteristics reinforce that treatment in diagnosed patients must be performed efficiently and regularly. Once the drug administration is not done efficiently, there is an intensification of several problems in the control of the disease, especially considering the Amazon region, which is an area with limited resources. This scenario may become even worse in the coming years, as more than 200,000 new leprosy cases have been confirmed worldwide only in 2018 [15].

Given this, this paper applies two multi-criteria decision-making models, *Analytic Hierarchy Process (AHP)* proposed by [16] and *ÉLimination et Choix Traduisant la REalité II (ELECTRE II)* proposed by [17] for prioritization in the process of drug administration in the treatment of Leprosy. This strategy is a low computational cost alternative that can be used throughout patient treatment with minimal operational effort and, can also mitigate the effects of not-so-efficient drug distribution. The results obtained are shown in an interactive data visualizer, which has several advantages in its use.

Data visualization has the potential to become an integral part of healthcare, as it provides multiple attributes in a single categories diagram or discrete state. These visualizations allow for comparison in different ways, which would help experts and decision makers [18]. In practice, with the previously configured analytical models, they can provide a controlling overview over a large data set, especially over the treatment and diagnosis of patients, which needs moderate clinical follow-up.

Considering that the treatment of Leprosy is a long-lasting and scarce process due to financial issues of the state, Multiple-Criteria Decision Method (MCDM) based on AHP-ELECTRE model were applied to make the care more efficient. The work uses a non-public database with patient information collected in the period 2015-2020 in 66 municipalities in the State of Para.

As the main contribution, this proposal seeks to provide an efficient mechanism to professionals for prioritizing and visualizing data of patients with greater severity, who have not yet been treated and should be evaluated in a prioritized manner.

The article is organized as follows: Section II presents the works related to the proposed theme. In section III, Materials and Methods are presented, containing the data set, AHP and ELECTRE II Method and the Case Study. In section IV the obtained results are shown and discussed. Finally, section VI presents the conclusion and a proposal for future work.

## Related work

Leprosy is one of the major public health problems in the world, imposing a reflection on its epidemiological situation and strategies for its confrontation. The disease continues to be one of the main causes of morbidity that, associated with sequelae, has affected in recent years about 2 million people worldwide [19]. Currently, there are several solutions that apply computational techniques to optimize the process of combating the disease, either in the sphere of diagnosis or treatment, in addition to other work fronts that seek a significant reduction in the prevalence of leprosy on a global level.

When it comes to multi-criteria decision methods applied to neglected diseases, there is a certain limitation in the scientific literature for problems of this nature. Despite this, the authors of [20] apply multi-criteria decision models for clinical trial analysis, evaluation of efficacy and safety of drug use for patients with *Huntington's* disease. Various aspects have been evaluated, such as the degree of importance of the drug, risk group, classification, and identification at the end.

Another discussion of great impact in the field of neglected diseases, is the shortage of drugs that involves life-threatening conditions for people. This approach is emphasized in [21], where the authors evaluate multicriteria decision strategies based on practical attributes of drug distribution. The study considers 37 strategies of different types based on expert opinion, and also applies an MCDM method, the AHP, in the decision-making process. The results indicated the main attributes that should be considered in the analysis of the drug chain and that should be prioritized in the experts' view.

Pharmaceutical supply chain involves different risk policies, from corporate spheres that may reach the pharmaceutical supply to the public health system. In this regard, [22] proposes a multi-criteria decision making methodology (MCDM) based on fuzzy-AHP approach to prioritize and rank risks in pharmaceutical supply chain. This study identifies 24 risks into five main risk measures through relevant literature and expert opinions, encompassing in its process, judgments and uncertainties involved in risk assessment. The results pointed out that pharmaceutical supply chain risks are the most important risks for industries in managing and reducing risk consequences.

Multi-criteria assessments are increasingly being employed in prioritizing health threats. In [23], the authors use *Multiple Criteria Decision Analysis (MCDA)* to determine weights for eleven criteria to prioritize non-critical COVID-19 patients for hospital admission in resource-limited healthcare environments. The method was applied in two main steps: specification of criteria for prioritizing COVID-19 patients (and levels within each criterion); and determination of weights for the criteria based on the knowledge and experience of experts in treating COVID-19 patients.

Another MCDA approach is presented by [24] who assess the value of a Multicriteria Decision Analysis (EVIDEM) framework for evaluation of Orphan drugs in Catalonia (Catalan Health Service). In the study, a method of weighting weights between criteria done through two techniques, with hierarchy and without hierarchy, was used. The results showed that the EVIDEM model is useful for evaluation of Orphan drugs; however, it can be improved based

on the adjustment of criteria, between weight and techniques. Furthermore, it was possible to identify which variables might be more important for Orphan drugs in Catalonia.

In [25], the authors use MCDA in establishing a multi-criteria decision-making model to evaluate dimensions (clinical, economic, and social) and criteria related to five target therapies for metastatic colorectal cancer. Three dimensions and nine criteria with respective target therapies in different dimensions are included in the study, in addition to a sensitivity analysis used to assess the robustness of the research results. In the evaluation, a questionnaire was applied to obtain information about the target therapies and the weights of the dimensions and criteria. The results showed not only the order of the weights and dimensions, but also the rankings of the value of the target therapies.

In [26] a new sensor-based method of disease symptom assessment is presented that can be applied in the neurological monitoring domain. The authors provide a quantitative approach for the recognition of symptoms and their intensity, which can be used for efficient and long-term planning of medication intake for patients with Parkinson's disease. Data were collected and used with a set of tests in order to verify the consistency of the proposed system and the implemented analysis methods.

In [27], the AHP is used to develop a multi-criteria model to evaluate the potential of various applications of Internet of Things (IoT) technologies in dementia care. Six IoT-based healthcare services were selected and compared with two conventional services (family health and assisted living facilities) in terms of effectiveness, safety, and patient perspectives. The results indicated a great potential of IoT technologies for problems of this nature, however, the importance of conventional dementia care services is still highly appreciated.

In [28] a heuristic method is proposed to select promising models based on their scores calculated in a multi-criteria scheme. The model ranking method considers the three main criteria: prediction error made by the model, correlation between target and model predictors, and model size. To avoid overfitting problems, models in the upper and lower categories were selected. The results obtained showed that the process of experimenting with the data outside the sample space, confirms the prediction accuracy of the ensemble scheme on data obtained from subtropical countries such as those in Asia, Africa and South America.

The rapid growth of the world population together with the challenges faced by public health intensifies the development of neglected diseases such as Leprosy. At the stage of disease detection for example, which is a sensitive period for health professionals, it is necessary to avoid inaccuracy during diagnosis, as this can lead to nerve damage and irreversible deformities for individuals. Therefore, it is remarkable that analytical models linked to multi-criteria decision methods, as presented above, can bring significant benefits to the public health sector, as well as to society.

Considering the great potential of analytical methods, such as MCDM for example, the use of these techniques in problems such as the one presented in this article is fundamental. Therefore, this work differs from others in that: (i) considering a predominant data set from the Amazon region, where there are limitations of health resources, problems with delays in diagnosis among others that make clinical follow-up difficult; (ii) presenting an analytical model to support the choice of individuals in order of priority, which is established according to the clinical status; (iii) a platform of data visualization for the long-term follow-up of patients and facilitating the distribution of drugs, which sometimes arrive seasonally.

## Material and methods

A multi-criteria approach has as characteristic several actors involved in a decision making process, having their own value judgment and recognizing the limits of objectivity, taking into

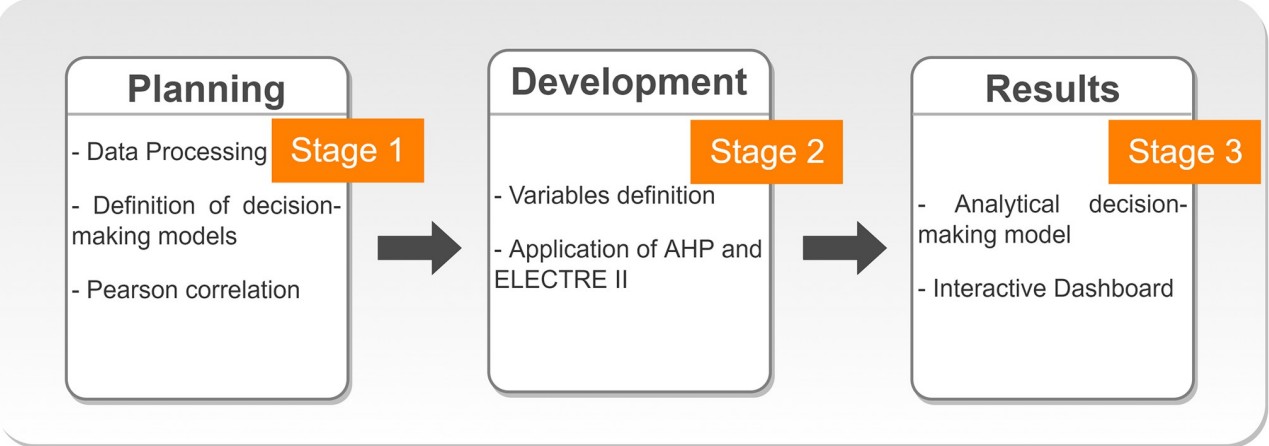

**Fig 1. Research methodology.** This figure presents the research methodology that was developed in the manuscript.

account their subjectivities [29]. A decision problem consists of a situation in which there are at least two alternatives, and this choice is conducted to meet various criteria. To build the model that will represent the problem being addressed, multi-criteria decision support models are used [30]. The proposed multi-criteria decision model was developed from a 3-step research methodology, as seen in Fig 1.

The research methodology used in this work consists of a sequence of activities divided into 3 steps (Fig 1). Initially, there is the planning stage, where data processing was performed with the application of Process Data Integration (PDI) for data treatment and manipulation, the definition of decision-making models and the calculation of Pearson's correlation index. In step 2, the variables were defined and applied in the analytical models. In step 3, the results obtained were analyzed and an interactive Dashboard was produced for data visualization and clinical follow-up of patients.

In this work, two analytical methods of MCDM were used, the AHP [31] and ELECTRE II [17], which besides having a high applicability in multi-criteria decision problems, obtained satisfactory results in the case study presented. For the choice of input attributes, the *Pearson Correlation Coefficient* [32] was calculated to estimate the degree of correlation between variables (divided into social, laboratory and neurological), verifying which variable exerts greater influence on the others as seen in Fig 2.

The attributes (Type of Patient, Patient Status, Number of Lesions, Clinical Form, Classification, Bacilloscopy and Treatment) used in Fig 2 characterize the epidemiological profile of patients in the pre- and post-diagnostic stages. From the illustration, it can be seen that the correlation between the variables Status and Type were those with the highest values (0.94), this is due to the fact that both attributes are directly linked to the diagnosis of patients, consequently having a moderate degree of importance during evaluation.

This study conforms to the Declaration of Helsinki and was approved by the Institute of Health Sciences Research Ethics Committee from Para Federal University (CAAE 26765414.0.0000.0018 CEP-ICS/UFPA). All individuals involved agreed voluntarily to participate and signed an informed consent form after receiving information about the study. Parents of minors or responsible adults signed their consent, allowing them to participate. All data analyzed were anonymized to protect the privacy of participants.

The correlation between classification and clinical form (0.81) is also noteworthy, which is justified by the fact that both are directly linked to the severity and amount of injuries in

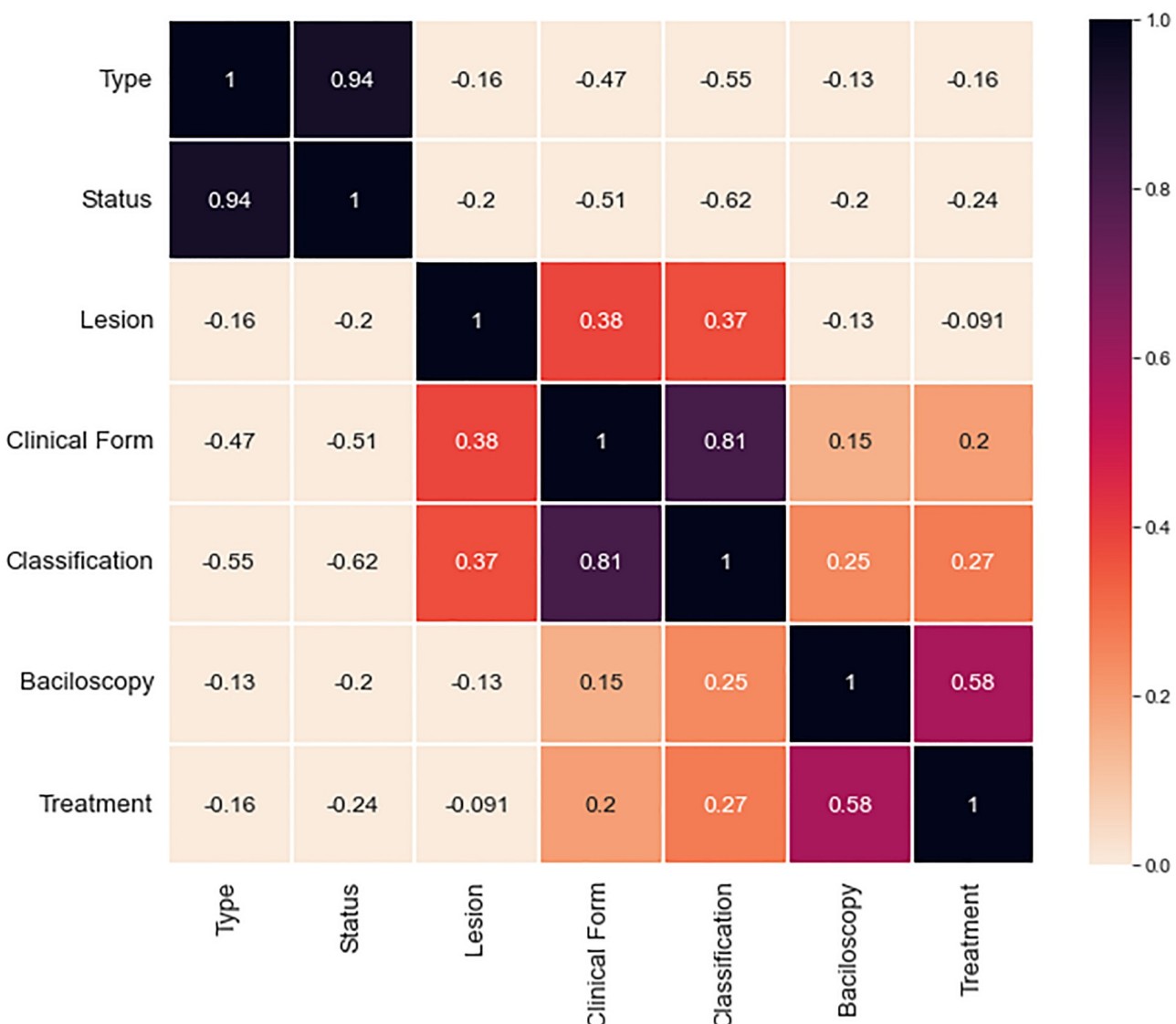

**Fig 2. Index of correlation between attributes.** This figure shows a Pearson correlation model that was applied across attributes this assessment.

individuals. The other variables present correlations of lesser magnitude for evaluation. Therefore, the application of Pearson's correlation helps to understand how a variable behaves in a scenario where another one is varying, thus it was possible to identify if there is any relationship between the variability of both.

## Dataset

The dataset used was extracted from a *Dataset* (Data is available at https://github.com/igorfalcao/hansysLeprosy) with non-public data obtained in the period 2015-2020 from patients in 66 municipalities in the state of Para, collected from this group's research project, which has diagnosed more than 637 cases and performed more than 4800 attendances in northern Brazil. The data involves personal information of patient care in clinical treatment. Access to data is allowed for the entire academic community that intends to contribute positively to

this research. This information refers to the clinical, laboratory, and neurological follow-up of people who are treated in public health units in partnership with other institutions, among them, the Federal University of Para. The information is collected through an Android platform in its raw form.

During the process, it was necessary to use techniques of *Data Science* to manipulate all the information, among them: (a) *Exploratory Data Analysis* (EDA), used to visualize the main characteristics of the set, identifying *Insights* and possible outputs, (b) *Extract, Transform, Load* (ETL) which is a data structuring process used to build infrastructures for easy access to information and (c) data visualization with the help of Business Intelligence (BI) tools.

One of the great practical benefits obtained in this visualization step is the interpretation of results from dynamic representations. For this data set, for example, a Dashboard was developed with the help of the Microsoft Power BI tool (Microsoft power bi. Available here: https://powerbi.microsoft.com/en-us, 2021) and PostgreSQL(PostgreSQL. Web resource: http://www.PostgreSQL.org/about, 1996), to show the results of the MCDM algorithms in relation to the profile of the evaluated individuals.

## AHP

The analytic hierarchy process, also known as the AHP algorithm, is a qualitative and quantitative decision analysis method. The method can model and quantify the decision-making thought process of complex systems. Using this approach, decision makers can divide previously complex problems into several layers and factors. After simple comparison and calculation of each factor, a variety of scheme weights can be obtained to provide a basis for the preferred scheme [31].

The weight distribution process is an essential step for multi-criteria models, since this action can directly influence decision making. However, the process is usually subjective and the weights reflect the imprecision and uncertainties of the decision-makers' judgments [33]. In practice, the attribution of weights between criteria indicates which attributes have greater influence during the decision-making process.

The basic principle of AHP is to evaluate the scheme according to the hierarchical structure (goal, criterion and condition). By comparing the above three items, the Eigenvalue of the judgment matrix is determined. The Eigenvector component is taken as the corresponding coefficient. In the matrix, criterion values are weighted to the judgments of expert weights [16]. In defining the aforementioned comparison matrix (pairwise), a scale is used to define the equal degree of relevance of one criterion to another. The scale has a range of levels from 1 to 9, where 1 equals equal equality and 9 extreme relevance equality.

In this work 7 criteria were considered (Fig 1) defined from the analysis of a *Dataset* consisting of a large amount of patient information. These attributes are extremely important to determine the degree of intensity of the disease and serve as a parameter for health professionals (physicians, nurses, physical therapists and technicians) to adjust the treatment.

The use of the AHP begins by decomposing the problem into a hierarchy of more easily analyzable criteria, as illustrated in Fig 3. After this moment, the decision makers systematically evaluate the alternatives by comparing them, two by two, within each one of the criteria. This determines the comparison matrix, or *Pairwise Comparison Matrix (PCM)*. To interpret and give the weights, it is necessary to normalize the previous comparative matrix. The matrix is normalized and the next step is initiated, where the calculation of the *Eigenvector* that will present the relative weights. The sum of the values in the vector determines the share or weight of that criterion.

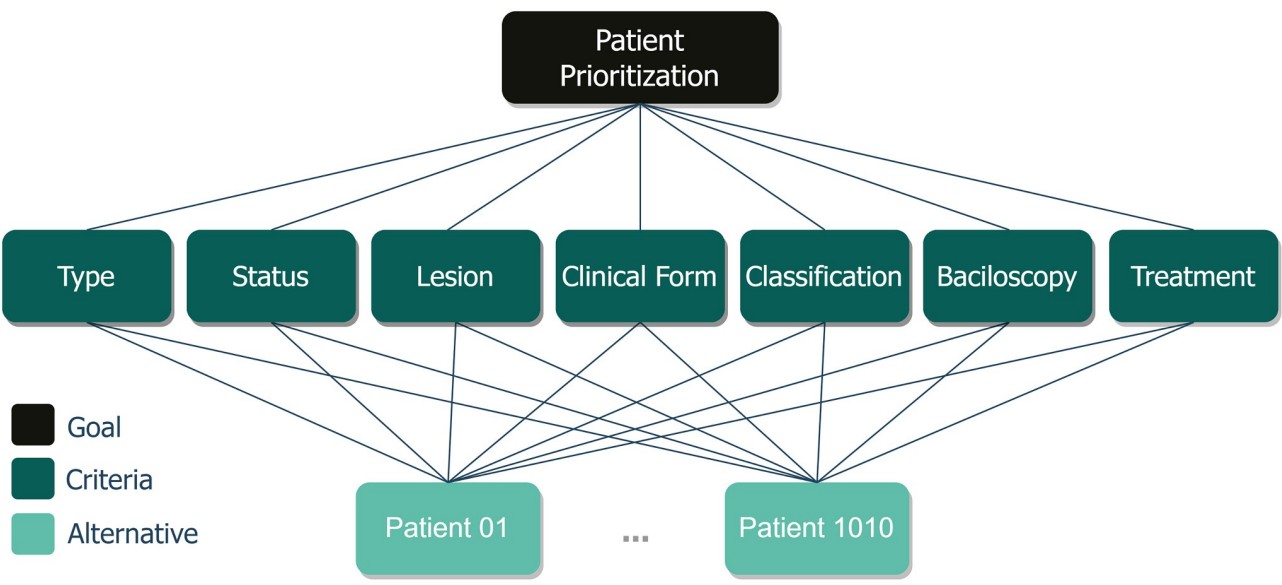

**Fig 3. Hierarchy of criteria/objectives.** The figure shows the basic hierarchical structure of the AHP method.

The next step of the process is to check the consistency of the data, where it is verified that the decision makers were consistent in their opinions. The next step is to calculate the main number of *Eigen* from the sum of the product of each element of the vector of *Eigen* by the total of the respective column of the original comparative matrix. Followed by the calculation of the *Consistency Index (CI)*, which is based on the main number of *5n*. The CI is obtained, according to [34].

$$CI = \frac{\lambda_{Max} - n}{n - 1} \qquad (1)$$

Where *CI* indicates the Consistency Index, the ($\lambda_{Max}$) equals the main number of *Eigen* and *n*, the number of criteria in the matrix, 7 in this case. To check whether the resulting value of *CI* is adequate for the problem, the *Consistency Ratio (CR)* was established by [34]. The matrix is considered consistent if the resulting value is less than 10% [35].

The value of *RI* can be observed through a table with fixed values used as reference and calculated in the laboratory, and are presented in Table 1. Then: the *CR* is determined by the ratio between the value of the consistency index and *Random Index (RI)*

$$CR = \frac{CI}{RI} < 0.1 \sim 10 \qquad (2)$$

The calculation of *CR* is given by Eq 2, using the values of *RI* and the numerical scale.

**Table 1. Random consistency indices.**

| N | 1 | 2 | 3 | 4 | 5 | 6 | 7 | 8 | 9 | 10 |
|---|---|---|---|---|---|---|---|---|---|---|
| RI | 0 | 0 | 0.58 | 0.9 | 1.12 | 1.24 | 1.32 | 1.41 | 1.45 | 1.49 |

This table shows Random consistency indices, which are values established by the Saaty scale.

## ELECTRE II

The ELECTRE II method, [36] and [37], aims to solve problems of ordering alternatives in the decision making process. This method uses concepts of concordance and discordance to indicate strong or weak over-ranking relationships among alternatives through criteria performance evaluation. The final ordering result is obtained through the process of distilling the alternatives with pre-orders built on the over-ranking relationships.

The concordance is related to how much one alternative is able to outperform another, based on the comparison between the best performances among the criteria. To calculate the concordance index, ELECTRE II considers the weights $w_j$ of the top performing criteria during the evaluation of two distinct alternatives $a$ and $b$, as shown in Eq 3. On the other hand, discordance addresses the relative disadvantage in performance $g_k$ of one alternative relative to the other. The discordance index can be obtained through the maximum difference in performance of the alternatives, as shown in Eq 4.

$$C(a, b) = \frac{1}{w} \sum_{j:g_j(a) \geq g_j(b)} w_j, \text{ where } w = \sum_{j=1}^{n} w_j \tag{3}$$

$$D(a, b) = \frac{\max_{1 \geq k \leq m} \quad g_k(b) - g_k(a)}{\max_{1 \geq k \leq m} \quad g_k(c) - g_k(d)} \tag{4}$$

To define the over ranking relations, ELECTRE II has concordance thresholds $c^+$, $c^0$ and $c^-$, and discordance thresholds $d^+$ and $d^-$. These thresholds are used in conjunction with the concordance and discordance indices between the alternatives, and are evaluated according to the conditions presented below:

- If $C(a, b) \geq c^+$ and $D(a, b) \leq d^+$ or $C(a, b) \geq c^0$ and $D(a, b) \leq d^-$, with $C(a, b) \geq C(b, a)$, then the alternative $a$ strongly overclasses $b$, and is denoted by $aS^F b$.

- If $C(a, b) \geq c^-$ and $D(a, b) \leq d^-$, with $C(a, b) \geq C(b, a)$, then the alternative $a$ weakly overclasses $b$, and is denoted by $aS^f b$.

After obtaining the indices of concordance and discordance between the alternatives, the distillation process is applied. In this sense, the partial ordering of the most adequate alternatives to the least adequate and the least adequate to the most adequate alternatives is performed, respectively, in descending and ascending distillations. For the ordering to be obtained a set $A$ that gathers all alternatives for decision making must be used.

Descending distillation occurs from the set A, deriving a subset $N$ of alternatives that are not strongly over ranked by other alternatives in A. With this, the subset $N$ is evaluated for the extraction of a subset $N'$, which includes the alternatives not weakly over ranked by other alternatives, that will occupy the first positions during ordering. This step of obtaining a subset $N'$ is repeated until all alternatives have been ordered.

Similarly, ascending distillation occurs from the set A, however, extracting a subset $Z$ of alternatives that are dominated and do not strongly overrank any other alternative from A. The first positions during ordering are occupied by a subset of alternatives $Z'$ that do not weakly overrank other alternatives from $Z$. The process of obtaining a subset $Z'$ is repeated until all alternatives have been ordered. Finally, the final ordering of the method is presented

in Eq 5.

$$\bar{v} = \frac{v' + v''}{2} \tag{5}$$

In Eq 5 it is performed the average of the value of the positions occupied by the same alternative in descending and ascending order. Thus, $v'$ and $v''$ represent the positions of an alternative $a$, respectively, in descending and ascending order, and $\bar{v}$ is the final ordering of $a$.

### Case study

This case study consists of analyzing the drug distribution scenario in the state of Para in the period 2015-2020. Two MCDM models (AHP and ELECTRE II) were applied to establish an order of priority in drug administration in Leprosy patients. The results are shown in a Dashboard that facilitates real-time monitoring and reporting. The work takes into account three factors: (a) the low quantity of medications distributed in the basic health units; (b) the in homogeneous distribution of medications; (c) patient abandoned the treatment.

The experiment considers the patient classification that is established by the WHO, divided into: New Case: People with a positive diagnosis for Leprosy; Relapse: patient who have already had the disease at some point in their lives; General: Patient who do not fit into even one of the other classes; Students: Patient in the middle of school. This distribution is seen in Fig 4, considering the precept of people in each class who did or did not receive some treatment. It is worth noting that only New Cases and Relapse are patients confirmed for Leprosy.

We evaluated 1010 patients distributed among the four types, as seen in Fig 4. For New Cases, only a small portion was attended, totaling 18%, and for Relapse, 43% were attended. In real numbers, these two groups of patients represent 616 people, with different physical and neurological characteristics. Considering only two groups of people Fig 5, patients as New Case and Non Patients as General, relapse and Student (people with no confirmed diagnosis), one can see that the amount of people who were attended is low, since in total, only 83% of the evaluated cases received some kind of treatment.

It is noted that there is a low quantity of people attended to in all scenarios, resulting in conditions with irreversible and multiresistant complications. In Fig 5, although the "patients" are clinically confirmed cases for Leprosy, there is no efficient control in drug distribution. This condition is a function established by professionals of the field (Doctors, Nurses, Physiotherapists, Teachers etc.), who take into consideration several clinical, laboratory and neurological attributes to define the treatment burden. Given this, the two MDCM models were applied based on the results of 7 attributes and the analyses made in Figs 4 and 5.

### Results

The results are shown from the execution of the AHP in 5 steps, starting with the construction of the decision hierarchy, pairwise evaluation of criteria defined by Type of Patient (A1), Patient Status (A2), Number of Lesions (A3), Clinical Form (A4), Classification (A5), Bacilloscopy (A6) and Treatment (A7). The next step is weight estimation, preference level definition and Global valuation of the alternatives, equivalent to one patient. The evaluation matrix was constructed and normalized, as seen in Table 2.

Besides the evaluation of the attributes (from A1 to A7) already normalized, the values resulting from the calculation of the *Eigenvector* and the weights obtained in the model steps are presented (Table 2). Subsequently, the values obtained with the Eigen calculation as well as the weights per criteria are displayed in Table 3. In the final stages of execution of the AHP

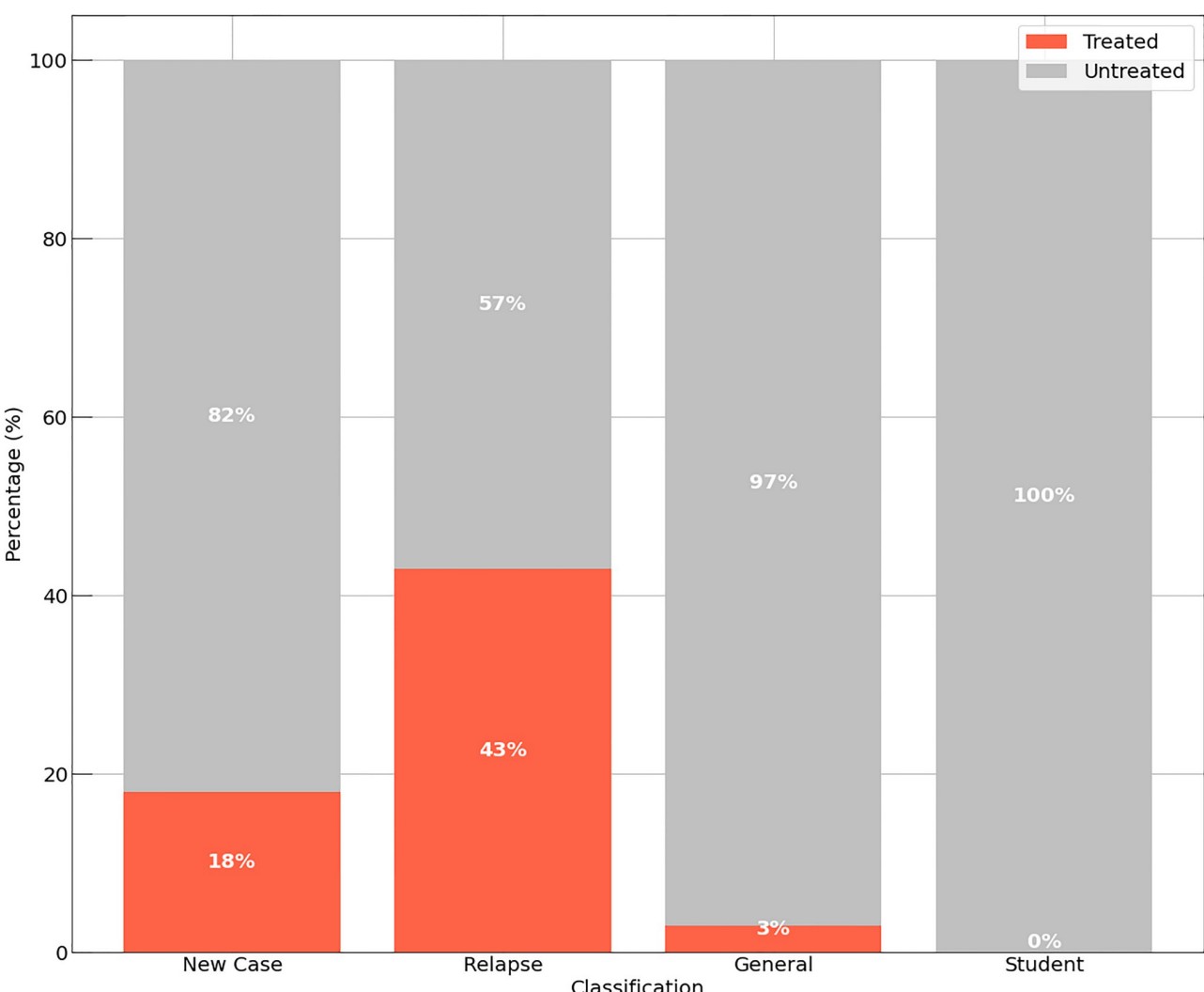

**Fig 4. Amount of people treated.** The figure shows the distribution of individuals in relation to their clinical classification.

Table 4. The obtained data were used as input data in ELECTRE II and consequently displayed in the data visualization step.

The AHP method obtained a consistency rate of 10.5%, a value already consolidated in the scientific literature as a favorable indication (Table 4). In ELECTRE II, the attributes (Type of Patient, Patient Status, Number of Lesions, Clinical Form, Classification, Bacilloscopy, Treatment) are used to select the alternatives that most need care, that is, patients with priority according to their clinical condition. The values of the AHP Comparative Matrix, used in the proposal for analysis of the collected criteria and definition of weights, are presented in Table 5.

In the final stages of execution of the AHP Table 4, the Main Value of Eigen and the consistency data of the models were calculated, values that are responsible for defining the degree of the consistency obtained (Fig 4). The obtained AHP Weights are values used in ELECTRE II Model steps below.

Although the fixed values of the AHP Weight Vector are reused, the values of the Input Matrix (Table 2) of the attributes will be used to apply ELECTRE II only in the initial stages. In

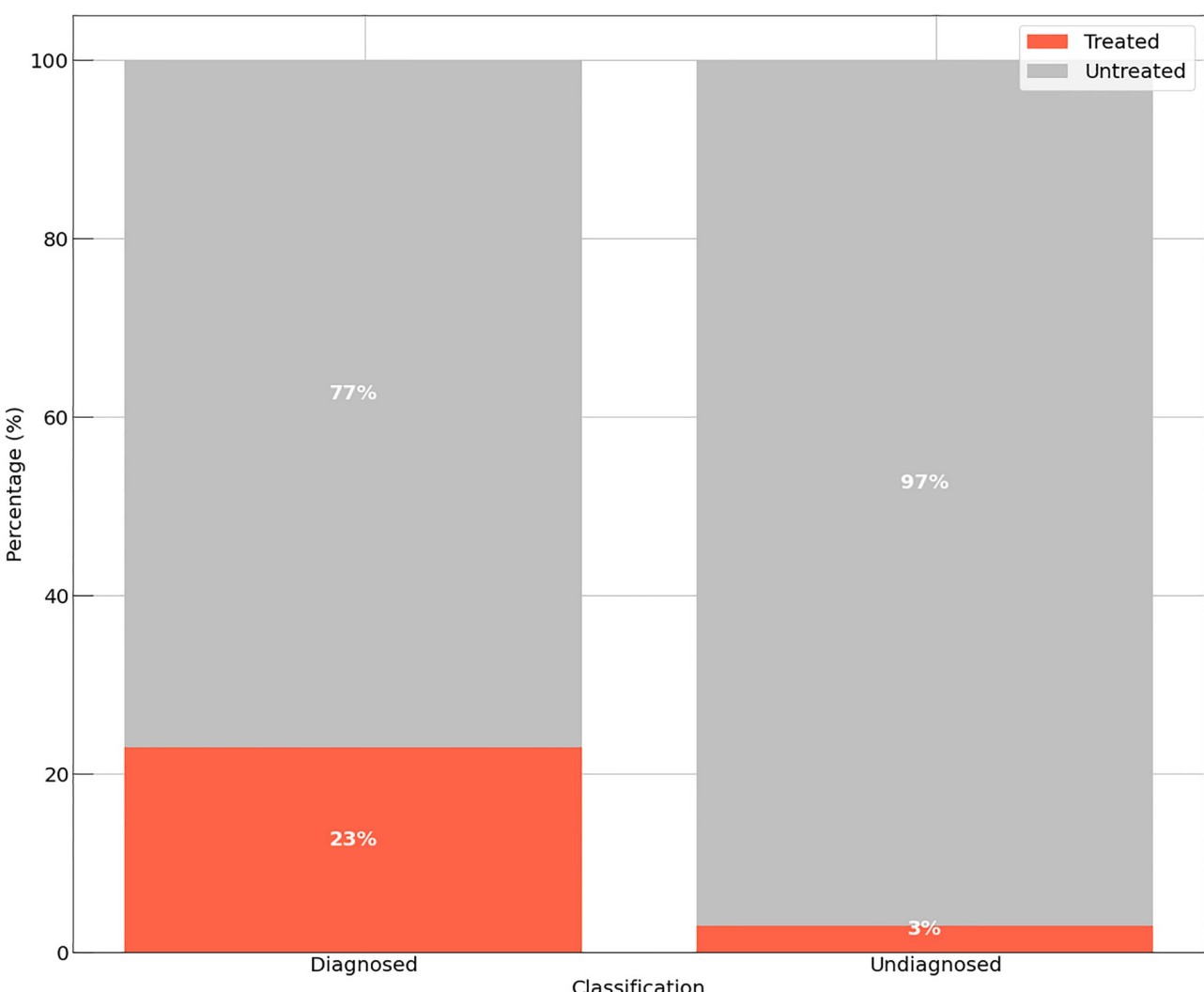

**Fig 5. Quantitative of patients.** The figure shows the percentages of diagnoses by individuals.

**Table 2. Peer to peer evaluation (normalized).**

| – | A1 | A2 | A3 | A4 | A5 | A6 | A7 |
|---|---|---|---|---|---|---|---|
| A1 | 1.00 | 0.13 | 0.17 | 0.33 | 0.33 | 0.50 | 0.50 |
| A2 | 8.00 | 1.00 | 8.00 | 6.00 | 6.00 | 6.00 | 5.00 |
| A3 | 6.00 | 0.13 | 1.00 | 4.00 | 4.00 | 6.00 | 8.00 |
| A4 | 3.00 | 0.17 | 0.25 | 1.00 | 2.00 | 3.00 | 6,00 |
| A5 | 3.00 | 0.17 | 0.25 | 0.50 | 1.00 | 3.00 | 3.00 |
| A6 | 2.00 | 0.17 | 0.17 | 0.33 | 0.33 | 1.00 | 4.00 |
| A7 | 2.00 | 0.20 | 0.13 | 0.17 | 0.33 | 0.25 | 1.00 |

This table shows the values obtained with the Peer to Peer Evaluation with criteria.

**Table 3. Calculation of eigenvalue and criteria weight.**

| Criteria | A1 | A2 | A3 | A4 | A5 | A6 | A7 |
|---|---|---|---|---|---|---|---|
| Eigenvector | 0.03 | 0.46 | 0.2 | 0.11 | 0.11 | 0.06 | 0.03 |
| Weight (w) | 0.02 | 0.507 | 0.185 | 0.095 | 0.102 | 0.052 | 0.029 |

This table presents the results obtained with the calculation of eigenvalue and criteria weight.

**Table 4. Consistency calculation.**

| Main Eigenvalue | Consistency Index (CI) | Consistency Ratio (CR) |
|---|---|---|
| 7.80 | 0.13 | 10.5% |

This table presents the results obtained with the consistency calculation of the AHP model.

Table 5 only 10 of 1010 alternatives of the model are expressed, where each alternative is equivalent to a patient with its physical and clinical characteristics.

Initially, the thresholds for applying ELECTRE II were defined, being $c^+ = 0.90$, $c^0 = 0.80$, $c^- = 0.75$, $d^+ = 0.20$ and $d^- = 0.10$. Then, the method was run to obtain the concordance and discordance indices according to the performances of the alternatives on the defined criteria. After this, the distillation process was performed to obtain the descending and ascending pre-orders. Finally, the average of the positions was obtained for each alternative, considering the different pre-orders, in order to obtain the final ranking. The positions associated with the final ordering of the alternatives are presented in Table 5.

Criteria A1 to A7, presented in Table 5, are categorical and have numerical representations according to the total number of categories. These representations were organized in a way that the higher the values of the criteria presented in the performance matrix, the higher the priority of the patient in the ordering process, due to the severity of his case. This is because, in this context, ELECTRE II aims to choose the alternatives that maximize the criteria performances, considering their weights.

**Table 5. Ranking of alternatives obtained with ELECTRE II.**

| Ranking ($\bar{v}$) | Alternatives | Criteria | | | | | | |
|---|---|---|---|---|---|---|---|---|
| | | A1 | A2 | A3 | A4 | A5 | A6 | A7 |
| 1° | P44 | 4 | 2 | 4 | 7 | 2 | 2 | 5 |
| 2° | P178 | 4 | 2 | 4 | 4 | 2 | 2 | 5 |
| 3° | P48 | 4 | 2 | 3 | 7 | 2 | 2 | 5 |
| 4° | P5 | 4 | 2 | 4 | 2 | 2 | 2 | 5 |
| 5° | P54 | 4 | 2 | 0 | 7 | 2 | 2 | 5 |
| 6° | P51 | 4 | 2 | 3 | 2 | 2 | 2 | 5 |
| 7° | P52 | 4 | 2 | 2 | 2 | 2 | 2 | 5 |
| 8° | P175 | 4 | 2 | 1 | 3 | 2 | 2 | 5 |
| 9° | P228 | 4 | 2 | 1 | 2 | 2 | 2 | 5 |
| 10° | P29 | 4 | 2 | 0 | 4 | 2 | 2 | 5 |

This table shows the result of final matrix with the ranking of alternatives obtained with the ELECTRE II model.

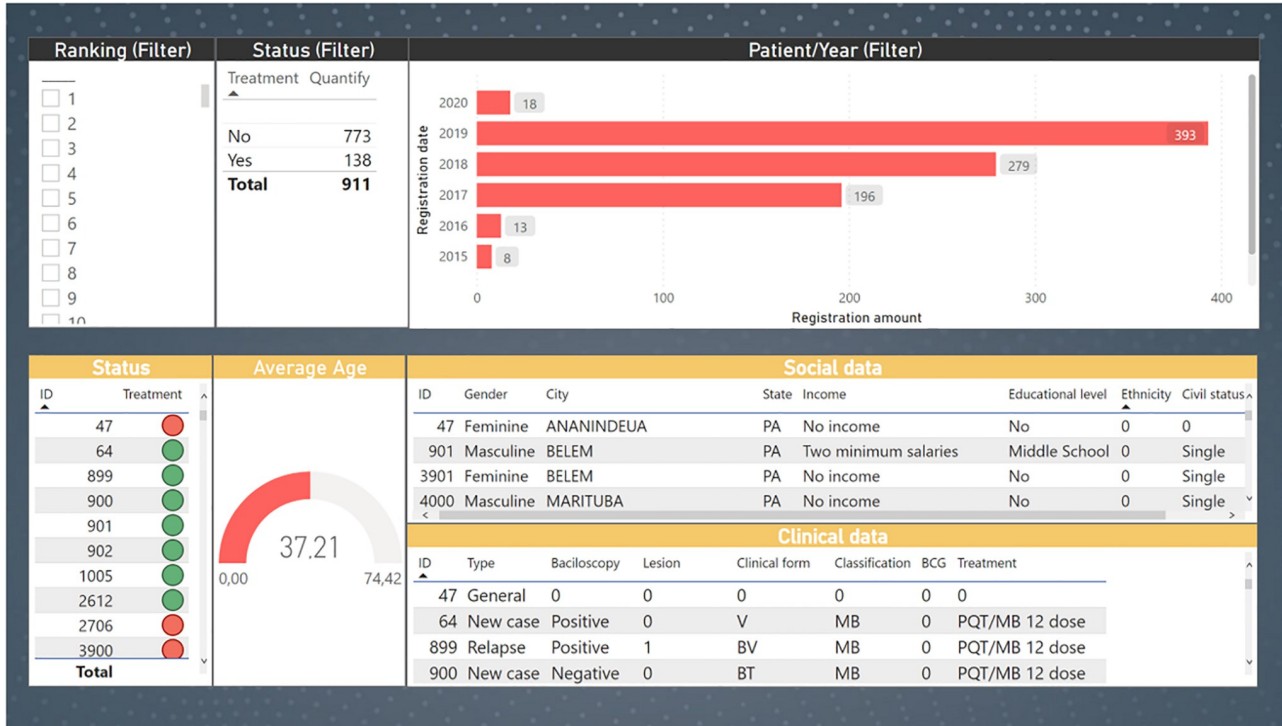

**Fig 6. Data visualization dashboard.** The figure shows an interactive dashboard designed for data visualization.

The priority alternatives are from the group of people diagnosed positive for Leprosy (New Case), which means that this public has an aggravated clinical condition, by their number of lesions, by the degree of each lesion and by all the variables used in the experiment. In addition, patients in the most severe stage of the disease have the highest values in each of the criteria evaluated, having received no treatment or an insufficient amount of medication for their case.

Necessarily, the model provides an optimized view of how an efficient drug administration is performed, considering attributes of greatest importance to the experts. This information is expressed in an interactive Dashboard (Fig 6) which is a solution with great benefits for public health institutions, because besides showing the results obtained in this work, it optimizes the long-term clinical follow-up of patients.

Fig 6 presents the Dashboard that shows the results obtained in two steps. The first (Fig 6A) displays the ELECTRE II selection filters on patient ranking, status and year of enrollment. The second (Fig 6B) displays the clinical outcomes of patients, social data and average age of patients that were selected in the previous step. The visualization tool developed offers significant benefits to specialists by optimizing the clinical follow-up process, as well as providing a method to help prioritize patients at the drug administration stage.

## Conclusion

This paper presented a MCDM model for prioritizing Leprosy patients during the drug administration step. The ranking results obtained with AHP, ELECTRE II, as well as clinical follow-up information are displayed in a Dashboard, which is an efficient solution for problems of this nature, given its accuracy and low computational cost. The experiment set up a ranking of patients who have a profile of greater severity in relation to the clinical picture of

the disease, concluding that individuals with a positive diagnosis, a number of lesions above 5, a positive result for bacilloscopy should be treated as a priority.

Patients without a diagnosis, with a low number of lesions and a negative bacilloscopy, appear at the end of the lists of alternatives. In practice, only 15% of the 911 patients evaluated received any treatment (either MDT or an alternative regimen), while 85% did not. Therefore, the prioritization process takes into account a number of physical, laboratory, and neurological factors, as well as the participation of specialists. In this scenario, the Dashboard was crucial for the evaluation, showing in an interactive and efficient way the whole clinical follow-up stage. On the other hand, for patients without treatment, it is also noted that the method obtained a gain in drug distribution of 68% of the 625 patients with positive diagnosis.

As future works, we intend to use a larger amount of attributes extracted from different Datasets, besides applying other MCDM methods to compare approaches and obtain an increasingly efficient process for medicine distribution. The Human-Computer Interaction (HCI) premises will also be included in the interactive data visualization stage, enabling the development of a more intuitive platform for the specialists.

## Acknowledgments

This work was supported by VALE S.A., and also by the National Council for Scientific and Technological Development (CNPq), Dean of Research and Graduate Studies (PROPESP) the Coordination for the Improvement of Higher Education Personnel. and by the Dean of Research and Graduate Studies (PROPESP) at the Federal University of Para.

## Author Contributions

**Conceptualization:** Karla T. F. Leite, Claudio G. Salgado, Moisés B. da Silva.

**Data curation:** Igor W. S. Falcão, Josafá G. Barreto.

**Formal analysis:** Igor W. S. Falcão, Marcos C. da R. Seruffo.

**Funding acquisition:** Karla T. F. Leite, Claudio G. Salgado, Adriano M. dos Santos.

**Investigation:** Igor W. S. Falcão, Harold D. de M., Junior, Patricia F. da Costa.

**Methodology:** Igor W. S. Falcão, Diego L. Cardoso, Moisés B. da Silva.

**Project administration:** Igor W. S. Falcão.

**Resources:** Claudio G. Salgado.

**Supervision:** Igor W. S. Falcão, Diego L. Cardoso, Karla T. F. Leite, Harold D. de M., Junior, Claudio G. Salgado, Moisés B. da Silva, Josafá G. Barreto, Patricia F. da Costa, Guilherme A. B. Conde, Marcos C. da R. Seruffo.

**Validation:** Igor W. S. Falcão, Daniel S. Souza, Diego L. Cardoso, Harold D. de M., Junior, Josafá G. Barreto, Patricia F. da Costa, Marcos C. da R. Seruffo.

**Visualization:** Igor W. S. Falcão, Guilherme A. B. Conde, Marcos C. da R. Seruffo.

**Writing – original draft:** Igor W. S. Falcão, Daniel S. Souza, Adriano M. dos Santos, Marcos C. da R. Seruffo.

**Writing – review & editing:** Igor W. S. Falcão, Daniel S. Souza, Diego L. Cardoso, Fernando A. R. Costa, Moisés B. da Silva, Josafá G. Barreto, Patricia F. da Costa, Adriano M. dos Santos, Marcos C. da R. Seruffo.

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
