## [Decision Letter · Decision Letter 0]

17 Jan 2022

PONE-D-21-36593Use of Multi-criteria Methods to Support Decision-Making in Drug Management for Leprosy PatientsPLOS ONE

Dear Dr. FALCÃO,

Thank you for submitting your manuscript to PLOS ONE. After careful consideration, we feel that it has merit but does not fully meet PLOS ONE’s publication criteria as it currently stands. Therefore, we invite you to submit a revised version of the manuscript that addresses the points raised during the review process.

We look forward to receiving your revised manuscript.

Kind regards,

Fausto Cavallaro, PhD

Academic Editor

PLOS ONE

Journal Requirements:

2.  Please include a complete ethics statement in the Methods section, including information on how the dataset was obtained, the name of the IRB and the approval number, and whether they approved the study or waived the need for approval. Please also clarify whether the participant provided consent, and if so, how, or whether the IRB waived the need for consent.

(The authors gratefully acknowledge the financial support from VALE Foundation, the Federal University of Para and Dean of Research and Graduate Studies and Coordination for the Improvement of Higher Education Personnel.)

(No, This Manuscript was developed with the help of the Vale Foundation, which represents the main organization promoting the project entitled Operational research and in-service training for leprosy hyperendemic areas in Maranhão and Para.)

6. We note that you have indicated that data from this study are available upon request. PLOS only allows data to be available upon request if there are legal or ethical restrictions on sharing data publicly. For more information on unacceptable data access restrictions, please see http://journals.plos.org/plosone/s/data-availability#loc-unacceptable-data-access-restrictions. 

Reviewers' comments:

Reviewer's Responses to Questions

**Comments to the Author**

1. Is the manuscript technically sound, and do the data support the conclusions?

Reviewer #1: Partly

Reviewer #2: Yes

2. Has the statistical analysis been performed appropriately and rigorously? 

Reviewer #1: N/A

Reviewer #2: Yes

3. Have the authors made all data underlying the findings in their manuscript fully available?

Reviewer #1: Yes

Reviewer #2: No

4. Is the manuscript presented in an intelligible fashion and written in standard English?

Reviewer #1: No

Reviewer #2: No

5. Review Comments to the Author

Reviewer #1: The main contribution of this paper is to use MCDM models (AHP [21] and TOPSIS [27]) to Support Decision-Making in Drug Management for Leprosy Patients through the study of analyzing the drug distribution scenario in the state of Par´a in the period 2016-2020. However, the paper in the present version is not acceptable. Because the research methodology that was developed in this work and many essential subjects are not clear, moreover many mistakes (typos, grammar errors, and others) can be found in it. Nevertheless, I have the feeling that the content may be of interest. But the paper should be better written and situate its results with respect to the literature.

Some comment is as follows:

1-The captions (description at figures) for figures are missing.

2-Explaining the contents of Table 1 better and writing the sentence ``The calculation of CR is given by Eq2.'' before Eq.2.

3-Explaining the prioritization patient process using (AHP) and (TOPSIS), i.e. illustrative of at least one case of a patient in the Case study.

4- Clarifying calculation of the Eigen value of the matrix, the corresponding Eigenvector, and Main Value, (Table 3 and Table 4).

5-There are several mistakes all over the text. Some of them have been corrected in the Manuscript Draft (see, attachment file) .

Reviewer #2: Dear editor of PLOS ONE,

The subject is "Use of Multi-criteria Methods to Support Decision-Making in Drug Management for Leprosy Patients"(PONE-D-21-36593), which is interesting and applicable. However, it will be suitable for publication if its shortcomings improve. In my view, the manuscript has several shortcomings as following;

1- In “Abstract”, the period of study stated as “… between 2015 and 2020”, but in the main body this state as “… 2016-2020”. Line 54, line 158, and line 262.

2- There is a series of superscript numbers and abbreviations in the text which must be modified in accordance with the format approved by the journal (e.g line 8, line 127, “…as per”).

3- The order of the references in the text is incorrect. For example, after the reference 8, the reference 10 comes, and after that, reference number 30, 26 and ...

4- Under the "Related Work", the authors stated FIVE related study (ref 13-17). Please discuss more study, For example, the following studies:

• Gilabert-Perramon, A., Torrent-Farnell, J., Catalan, A., Prat, A., Fontanet, M., Puig-Peiró, R., . . . Badia, X. (2017). “Drug evaluation and decision making in catalonia: development and validation of a methodological framework based on multi-criteria decision analysis (MCDA) for orphan drugs”. International Journal of Technology Assessment in Health Care, 33(1), 111-120. doi:10.1017/S0266462317000149

• A Moosivand, M Rangchian, L Zarei, F Peiravian, G Mehralian, ... (2021). “An application of multi-criteria decision-making approach to sustainable drug shortages management: evidence from a developing country”. Journal of Pharmaceutical Health Care and Sciences 7 (1), 1-11

• Vinayak Vishwakarma, Chandra Prakash, Mukesh Kumar Barua.(2016).” A fuzzy-based multi criteria decision making approach for supply chain risk assessment in Indian pharmaceutical industry”. International Journal of Logistics Systems and Management. Vol. 25, No. 2. pp 245-265

• Jason C. Hsu , Jia-Yu Lin, Peng-Chan Lin, Yang-Cheng Lee (2019). “Comprehensive value assessment of drugs using a multi-criteria decision analysis: An example of targeted therapies for metastatic colorectal cancer treatment”. PLOS ONE. https://doi.org/10.1371/journal.pone.0225938.

5- The figure’s number in the manuscript and PDF provided for review does not match.

6- There are occasional minor typos in the manuscript. For example, “in homogeneous” instead to “inhomogeneous”.

7- In line 265-266, the authors stated that “…three factors: (a) the low quantity of medications distributed in the basic health units; (b) the in homogeneous distribution of medications; (c) patient evasion.” I did not understand the concept of "patient evasion ". I think here there is a need to add an explanation.

8- What is the meaning of "the responsible agencies" in line 269?

9- In line 273-274, it is stated that “It is worth noting that 273 only New Cases are patients confirmed for Leprosy”, which does not conform to the definition provided for “Relapse” in line 270.

10- The patient classification must be the same throughout the manuscript. At present, for example, for “Relapse“ in the text and figures, similar words such as recurrence etc. have been used, which need to be uniform.

11- The line 275 to 282 does not make sense. The numbers and percentages provided do not match. The sum of 18% and 43% is equal to 61%, which is equal to 616 out of 1010 patients.

12- The authors have stated in line 286-288 that “This condition is a function established by professionals of the field, who take into consideration several clinical, laboratory and neurological attributes to define the treatment burden.” Please provide more information about the composition and number of experts participating in the study.

13- It is expected that few of the readers of this journal are familiar with the analysis method applied by the authors. In order to make it easier for novice readers to understand the contents, please provide additional explanations on the meaning of the weights in the tables and the interpretation of the results (numerical values).

14- The authors do not discuss the implications of their findings. Doing so would make the study more interesting to the reader and actually support potential policy changes. I suggest starting the discussion by repeating the objective of the study and followed by each main finding that answered each specific research questions.

15- Overall the discussion section need to be added, and must be written exactly.

6. PLOS authors have the option to publish the peer review history of their article (what does this mean?). If published, this will include your full peer review and any attached files.

Reviewer #1: No

Reviewer #2: No

---

## [Author Response · Author response to Decision Letter 0]

9 Sep 2022

Dear Editor,

We thank again the evaluations of the two reviewers regarding the article “Use of Multi-criteria Methods to Support Decision-Making in Drug Management for Leprosy Patients”. Thus, after evaluating what was indicated by the reviewers, several pertinent points were considered, which led us to a profound restructuring of the manuscript, aiming to make it even more coherent and with better results. Thus, we highlight the following points:

-The use of the analytical model AHP and TOPSIS, which was used in the initial version of the article, was rethought. After testing with other models, it was found that the hybrid use of AHP with ELECTRE II (instead of TOPSIS) generated more reliable results for leprosy drug decision making. Therefore, the new version of the paper brings results from the combination of AHP and ELECTRE II.

-With the aforementioned change in the decision model, more accurate results were obtained, thus, it was necessary to restructure the sections of the article, so the TOPSIS section was renamed as ELECTRE II (Abstract, Introduction, Related Work, Material and Methods, Case Study, Results, Conclusion).

-The title of the article "Use of Multi-criteria Methods to Support Decision-Making in Drug Management for Leprosy Patients" is consistent with the proposal, even after the modifications suggested by the reviewers and the change in the analytical model. However, the title "A Study About Management of Drugs for Leprosy Patients Under Medical Monitoring: A Solution Based on AHP-Electre Decision-Making Methods" makes the solution of the study even more evident. 

-With the amount of adjustments mentioned above, added with the indication of the Reviewers, comments, of English revision, it was necessary the invitation of one more author ADRIANO MADUREIRA DOS SANTOS, who was responsible for generating new results and performing a complete revision of the English writing. Therefore, this author was added to the list of authors.

The authors of the paper (PONE-D-21-36593R1) would like to express their gratitude for the valuable comments and advice of the reviewers; we believe that these changes have improved the quality and readability of the paper.

The captions (description at figures) for figures are missing.

Authors’ response: The manuscript has been adjusted, adding the correct description and caption for the figures.

Explaining the contents of Table 1 better and writing the sentence ``The calculation of CR is given by Eq2.'' before Eq.2.

Authors’ response: The description in table 1 of the AHP section has been adjusted as requested. (Material and Methods section, AHP subsection, page 7)

Explaining the prioritization patient process using (AHP) and (TOPSIS), i.e. illustrative of at least one case of a patient in the Case study.

Authors’ response: All the items mentioned have been corrected, illustrating the patient examples in the case study (Material and Methods section, Case Study subsection, page 8).

Clarifying calculation of the Eigen value of the matrix, the corresponding Eigenvector, and Main Value, (Table 3 and Table 4).

Authors’ response: The table 3 and 4 corresponding Eigenvector items were correctly adjusted. (Results section, page 10). 

There are several mistakes all over the text. Some of them have been corrected in the Manuscript Draft (see, attachment file).

Authors’ response: We have carried out an extensive spelling check, removing redundancies and typos throughout the text.

In “Abstract”, the period of study stated as “… between 2015 and 2020”, but in the main body this state as “… 2016-2020”. Line 54, line 158, and line 262.

Authors’ response: The cited excerpt from the abstract was adjusted according to the request (abstract, page 1).

There is a series of superscript numbers and abbreviations in the text which must be modified in accordance with the format approved by the journal (e.g line 8, line 127, “…as per”). 

Authors’ response: We have carried out an extensive spelling check, correcting the mentioned items and checking the journal format.

The order of the references in the text is incorrect. For example, after the reference 8, the reference 10 comes, and after that, reference number 30, 26 and ...

Authors’ response: New references have been added to the manuscript. Thereon, the order of the references were adjusted according to the request. 

Under the "Related Work", the authors stated FIVE related study (ref 13-17). Please discuss more study, For example, the following studies:

Authors’ response: All references recommended by the reviewers were added in the related work section of the manuscript (Related work, page 3).

-Gilabert-Perramon, A., Torrent-Farnell, J., Catalan, A., Prat, A., Fontanet, M., Puig-Peiró, R., . . . Badia, X. (2017). “Drug evaluation and decision making in catalonia: development and validation of a methodological framework based on multi-criteria decision analysis (MCDA) for orphan drugs”. International Journal of Technology Assessment in Health Care, 33(1), 111-120. doi:10.1017/S0266462317000149.

-A Moosivand, M Rangchian, L Zarei, F Peiravian, G Mehralian, ... (2021). “An application of multi-criteria decision-making approach to sustainable drug shortages management: evidence from a developing country”. Journal of Pharmaceutical Health Care and Sciences 7 (1), 1-11.

-Vinayak Vishwakarma, Chandra Prakash, Mukesh Kumar Barua.(2016).”A fuzzy-based multi criteria decision making approach for supply chain risk assessment in Indian pharmaceutical industry”. International Journal of Logistics Systems and Management. Vol. 25, No. 2. pp 245-265.

-Jason C. Hsu , Jia-Yu Lin, Peng-Chan Lin, Yang-Cheng Lee (2019). “Comprehensive value assessment of drugs using a multi-criteria decision analysis: An example of targeted therapies for metastatic colorectal cancer treatment”. PLOS ONE. https://doi.org/10.1371/journal.pone.0225938.

The figure’s number in the manuscript and PDF provided for review does not match.

Authors’ response: The figure’s number order in the PDF format in the manuscript were adjusted according to the review. 

There are occasional minor typos in the manuscript. For example, “in homogeneous” instead to “inhomogeneous”.

Authors’ response: We have carried out an extensive spelling check, removing redundancies and typos throughout the text.

In line 265-266, the authors stated that “…three factors: (a) the low quantity of medications distributed in the basic health units; (b) the in homogeneous distribution of medications; (c) patient evasion.” I did not understand the concept of "patient evasion ". I think here there is a need to add an explanation.

Authors’ response: The cited sentence was adjusted in the manuscript, improving the explanation of items such as "patient evasion" that can confuse the reader (Material and 

Methods section, Case Study subsection, page 8).

What is the meaning of "the responsible agencies" in line 269?

Authors’ response: The mentioned expression was removed, with the aim of improving the understanding of the manuscript. 

In line 273-274, it is stated that “It is worth noting that 273 only New Cases are patients confirmed for Leprosy”, which does not conform to the definition provided for “Relapse” in line 270.

Authors’ response: The manuscript has been adjusted, correcting the amount expressed in the text relapse cases (Results section, page 9). 

The patient classification must be the same throughout the manuscript. At present, for example, for “Relapse“ in the text and figures, similar words such as recurrence etc. have been used, which need to be uniform.

Authors’ response: The manuscript has been adjusted, meeting all reviews in relation to the items mentioned (Material and Methods section, Case Study subsection, page 9). 

The line 275 to 282 does not make sense. The numbers and percentages provided do not match. The sum of 18% and 43% is equal to 61%, which is equal to 616 out of 1010 patients.

Authors’ response: The percentage of new cases and relapse was adjusted in the manuscript, meeting the requested correlation (Material and Methods section, Case Study subsection, page 9).

The authors have stated in line 286-288 that “This condition is a function established by professionals of the field, who take into consideration several clinical, laboratory and neurological attributes to define the treatment burden.” Please provide more information about the composition and number of experts participating in the study.

Authors’ response: An explanation of the types of experts and people who participated in the study was added to the manuscript. This condition improved the explanation of the 

scenarios and the attributes of the experiment's inputs (Material and Methods section, Case Study subsection, page 9).

It is expected that few of the readers of this journal are familiar with the analysis method applied by the authors. In order to make it easier for novice readers to understand the contents, please provide additional explanations on the meaning of the weights in the tables and the interpretation of the results (numerical values).

Authors’ response: A new explanation about the meaning of the weights in the tables has been added to results, in order to facilitate the explanation of the input data in the 

manuscript (Results section, page 10).

The authors do not discuss the implications of their findings. Doing so would make the study more interesting to the reader and actually support potential policy changes. I suggest starting the discussion by repeating the objective of the study and followed by each main finding that answered each specific research questions.

Authors’ response: The text underwent an extensive spelling check, where new items were added in its structure with the aim of improving its presentation and its discussion of the contributions.

Overall the discussion section need to be added, and must be written exactly.

Authors’ response: The text underwent an extensive spelling check. The manuscript underwent an extensive revision in its structure, where new elements were added and a discussion was made of certain of the main contributions of the text, as the benefits of a study of this nature for the academy and on the social impact of the study

We are grateful to the reviewers for all their advice and will be very pleased to add any further information if this is required.

Thank you for your kind attention.

Yours sincerely,

---

## [Decision Letter · Decision Letter 1]

10 Oct 2022

A Study About Management of Drugs for Leprosy Patients Under Medical Monitoring: A Solution Based on AHP-Electre Decision-Making Methods

PONE-D-21-36593R1

Dear Dr. FALCÃO,

We’re pleased to inform you that your manuscript has been judged scientifically suitable for publication and will be formally accepted for publication once it meets all outstanding technical requirements.

Kind regards,

Fausto Cavallaro, PhD

Academic Editor

PLOS ONE

Additional Editor Comments: The authors addressed the reviewers comments. The paper can be accepted.

Reviewers' comments:

Reviewer's Responses to Questions

**Comments to the Author**

1. If the authors have adequately addressed your comments raised in a previous round of review and you feel that this manuscript is now acceptable for publication, you may indicate that here to bypass the “Comments to the Author” section, enter your conflict of interest statement in the “Confidential to Editor” section, and submit your "Accept" recommendation.

Reviewer #1: (No Response)

2. Is the manuscript technically sound, and do the data support the conclusions?

Reviewer #1: (No Response)

3. Has the statistical analysis been performed appropriately and rigorously? 

Reviewer #1: (No Response)

4. Have the authors made all data underlying the findings in their manuscript fully available?

Reviewer #1: (No Response)

5. Is the manuscript presented in an intelligible fashion and written in standard English?

Reviewer #1: (No Response)

6. Review Comments to the Author

Reviewer #1: (No Response)

7. PLOS authors have the option to publish the peer review history of their article (what does this mean?). If published, this will include your full peer review and any attached files.

Reviewer #1: No

---

## [Editor Report · Acceptance letter]

8 Nov 2022

PONE-D-21-36593R1 

A Study About Management of Drugs for Leprosy Patients Under Medical Monitoring: A Solution Based on AHP-Electre Decision-Making Methods 

Dear Dr. Falcão:

I'm pleased to inform you that your manuscript has been deemed suitable for publication in PLOS ONE. Congratulations! Your manuscript is now with our production department. 

Kind regards, 

on behalf of

Professor Fausto Cavallaro 

Academic Editor

PLOS ONE